# TAMING THE FORENSIC SINGULARITY:
# A REGULARIZED HYPERBOLIC FRAMEWORK FOR GENERALIZABLE AI-GENERATED IMAGE DETECTION

## ABSTRACT

Detecting AI-generated images is a critical task in multimedia forensics, yet the generalization of detectors to unseen generative models remains a persistent challenge. While pre-trained Vision Transformers (ViTs) have emerged as powerful feature extractors, existing forensic methods often default to using final-layer features, which may discard crucial forensic traces. We embark on a systematic probe into the latent representations of various ViTs and uncover a universal phenomenon we term the "Forensic Singularity": a narrow region within the ViTs' mid-level layers where forensic separability culminates before giving way to semantic abstraction. To harness the immense potential of this "singularity" layer while mitigating its high risk of overfitting to limited training data, we propose a Regularized Hyperbolic Framework. Our framework learns a polar representation in the Poincaré Ball by disentangling semantic content and forensic evidence: the final-layer semantic feature guides the direction, while the singularity-layer forensic feature determines the radius. This inherent geometric constraint regularizes the model, promoting a more generalizable decision boundary. Extensive experiments demonstrate that our approach not only establishes a new state of the art on multiple datasets, and exhibits superior generalization performance on unseen generative models. Our work provides both a powerful new tool for AI forensics and a deeper insight into how the hierarchical representations of ViTs can be effectively harnessed.

## 1 INTRODUCTION

The last decade has witnessed a meteoric rise in the capabilities of generative artificial intelligence (GenAI). Successive waves of innovation, from Variational Autoencoders Kingma & Welling (2013) and Generative Adversarial Networks Goodfellow et al. (2020) to more recent Diffusion Models Song et al. (2020)and Flow-based Lipman et al. (2022), have progressively narrowed the gap between synthetic and real-world imagery. Today's models can produce high-fidelity, photorealistic images that are often indistinguishable to the human eye, democratizing content creation but also opening the Pandora's Box to widespread malicious use, including misinformation campaigns, social engineering, and the creation of non-consensual explicit content. This escalating threat has established AI-generated image detection as a critical and urgent field of research.

A primary challenge in AI forensics is **generalization**. A detector trained on images from one generative model often fails catastrophically when tested against images from a new, unseen architecture. To combat this, a prevailing paradigm has shifted towards leveraging large-scale, pre-trained Vision Transformers (ViTs) Dosovitskiy et al. (2020) as universal feature extractors. Pioneering works demonstrated that frozen features from vision-language models like CLIP possess a remarkable ability to capture subtle, generalizable artifacts, forming a strong baseline for universal fake image detection Ojha et al. (2023); Cozzolino et al. (2024). Subsequent research Yan et al. (2024b); Chen et al. (2024); Zhou et al. (2025) has focused on sophisticated fine-tuning techniques to further adapt these CLIP-based backbones to the forensic task.

A key observation is that a common thread unites these successful approaches: they almost exclusively rely on the feature representation from the **final layer** of the ViT encoder, assuming that the

most semantically abstract features are optimal. We argue that this assumption may be fundamentally flawed for AI forensics. Recent breakthroughs in representation learning have revealed that for many vision tasks beyond pure classification, the optimal features reside not at the network's output, but in its **mid-level layers** Bolya et al. (2025). This suggests that the very process of semantic abstraction may actively discard the fine-grained textural and structural details that constitute the most crucial forensic traces.

This raises a critical, yet largely unexplored, question: *Where in a Vision Transformer does the most potent forensic signal reside?* Our work addresses this fundamental gap. We posit that by first identifying the optimal intrinsic feature source, we can build more powerful and robust detectors without necessarily resorting to complex fine-tuning. Our contributions are threefold:

- We conduct a large-scale, systematic probe into the layer-wise representations of seven ViT models across three diverse datasets. Our analysis uncovers a universal phenomenon we term the **"Forensic Singularity"**—a consistent peak in forensic separability located in the models' mid-level layers.

- We reveal that this singularity is a double-edged sword, offering immense discriminative potential but also a high risk of overfitting. To resolve this, we propose a novel, theoretically-grounded **Regularized Hyperbolic Framework** that tames the singularity features by leveraging the unique geometry of hyperbolic space.

- Our proposed method, guided by these findings, is shown to achieve state-of-the-art performance and, more importantly, to demonstrate significantly superior generalization to unseen generative models.

## 2 RELATED WORKS

**AI-Generated Image Detection.** The challenge in AI-generated image (AIGI) detection is generalizing to unseen generative models. A recent paradigm shift has established large-scale Vision-Language Models (VLMs) like CLIP as powerful backbones for this task. Initial works demonstrated that even frozen CLIP features, with simple classifiers, offer surprising generalization capabilities Ojha et al. (2023); Cozzolino et al. (2024). This has spurred a wave of research into more sophisticated adaptation and fine-tuning methods that build upon the CLIP encoder, utilizing techniques such as prompt tuning Khan & Dang-Nguyen (2024), orthogonal subspace decomposition Yan et al. (2024b), or on-manifold adversarial training Zhou et al. (2025). Concurrently, another line of work reframes the problem as an anomaly detection task, where the goal is to model the manifold of real images and detect forgeries as out-of-distribution samples. This has been approached through specialized loss functions that focus only on artifacts Rajan & Lee (2025) or by training on real images exclusively Bi et al. (2023). Our work aligns with this anomaly detection philosophy but realizes it at a geometric level. We leverage the inherent asymmetric geometry of hyperbolic space as a structural prior to distinguish the singular "real" manifold from the diverse "generated" manifold.

**Analysis of Network Representations.** Our methodology is inspired by the growing body of literature on probing the internal representations of deep networks to find optimal features for downstream tasks. A landmark study by Bolya et al. (2025) systematically demonstrated that for a wide range of perceptual tasks in the visual domain, the most effective representations reside in the **mid-level layers** of a network, not its final output. This principle of leveraging latent-space features is gaining traction across modalities. For instance, Mitra et al. (2025) showed that features from sparse attention heads within Large Multimodal Models (LMMs) significantly outperform final-layer features for few-shot vision-language classification. Furthermore, a parallel phenomenon has been observed in the audio domain, where Kheir et al. (2025) found that lower-to-mid layers of self-supervised models provide the most discriminative features for audio deepfake detection. While the superiority of mid-level features is becoming evident in various domains, to our knowledge, our work is the first to conduct such a **systematic, cross-model, and cross-dataset layer-wise analysis specifically for the task of AI-generated image forensics**. The discovery of the "Forensic Singularity"—the precise locus of maximal forensic signal—extends these prior findings. It not only confirms the validity of this principle in the forensic context but also provides a quantitative,

data-driven foundation for a new generation of detectors that are explicitly designed to harness the power of these potent mid-level features.

## 3 LAYER-WISE SEPARABILITY ANALYSIS

**Motivation:** Building upon the insights from the related work, particularly the demonstrated superiority of mid-level features for various perceptual tasks Bolya et al. (2025), we challenge the conventional reliance on final-layer features in AI forensics. We hypothesize that the optimal, intrinsic forensic signal resides within the intermediate layers of a Vision Transformer, where crucial non-semantic artifacts have not yet been discarded by semantic abstraction. This motivates our work's foundational first step: to move beyond adaptation of final-layer features and instead discover the most potent feature source. To achieve this in a systematic and classifier-agnostic manner, we introduce our layer-wise separability analysis as a quantitative probe.

### 3.1 LAYER-WISE SEPARABILITY AS A FORENSIC PROBE

To systematically answer the question of where the most potent forensic signal resides, we require a method to quantify the discriminative power of features at each layer of a ViT. Critically, this method should be **classifier-agnostic**, meaning it should measure the intrinsic separability of the feature space itself, rather than the performance of a specific downstream classifier which might introduce its own biases. To this end, we propose using a **layer-wise separability score**, which acts as our Euclidean probe into the model's latent representations. The core intuition behind our separability score is geometric: a feature representation is highly effective for forensics if it maps real images to a compact, coherent cluster in the latent space, while simultaneously mapping generated images to another distinct and distant cluster. We formalize this intuition by defining separability as the ratio of the between-class distance to the within-class distance.

**Formal Definition.** Let $\mathcal{M}$ be a ViT with $L$ layers. For a dataset of real images $\mathbb{D}_{\text{real}}$ and generated images $\mathbb{D}_{\text{fake}}$, we first extract the set of feature vectors from layer $l$, denoted as $\mathbb{Z}_{l,\text{real}}$ and $\mathbb{Z}_{l,\text{fake}}$ respectively. The features $\boldsymbol{z} \in \mathbb{R}^d$ are derived by averaging the patch tokens from the layer's output for each image.

First, we compute the **between-class distance**, $d_B(l)$, which measures the distance between the class centroids. A larger distance implies greater separation on average. We define it using the L2 norm ($L^2$):

$$d_B(l) = \|\boldsymbol{\mu}_{l,\text{real}} - \boldsymbol{\mu}_{l,\text{fake}}\| \tag{1}$$

where $\boldsymbol{\mu}_{l,\text{real}} = \frac{1}{|\mathbb{Z}_{l,\text{real}}|} \sum_{\boldsymbol{z} \in \mathbb{Z}_{l,\text{real}}} \boldsymbol{z}$ and $\boldsymbol{\mu}_{l,\text{fake}} = \frac{1}{|\mathbb{Z}_{l,\text{fake}}|} \sum_{\boldsymbol{z} \in \mathbb{Z}_{l,\text{fake}}} \boldsymbol{z}$ are the centroids of the real and fake classes.

Next, we compute the **within-class distance**, $d_W(l)$, which measures the overall compactness of the clusters. We compute the mean intra-class distance for each class. For the real class, this quantity, denoted $s_{l,\text{real}}$, is the average L2 distance of each feature to its centroid: $s_{l,\text{real}} = \frac{1}{|\mathbb{Z}_{l,\text{real}}|} \sum_{\boldsymbol{z} \in \mathbb{Z}_{l,\text{real}}} \|\boldsymbol{z} - \boldsymbol{\mu}_{l,\text{real}}\|$. Analogously, the fake class distance is $s_{l,\text{fake}} = \frac{1}{|\mathbb{Z}_{l,\text{fake}}|} \sum_{\boldsymbol{z} \in \mathbb{Z}_{l,\text{fake}}} \|\boldsymbol{z} - \boldsymbol{\mu}_{l,\text{fake}}\|$.

The final within-class distance, $d_W(l)$, is then defined as the average of these two dispersion values:

$$d_W(l) = \frac{s_{l,\text{real}} + s_{l,\text{fake}}}{2} \tag{2}$$

Finally, the **layer-wise separability score**, $S(l)$, for layer $l$ is defined as the ratio of these two quantities. A small stabilization term $\varepsilon$ is added to the denominator to prevent division by zero:

$$S(l) = \frac{d_B(l)}{d_W(l) + \varepsilon} \tag{3}$$

This score provides a normalized measure of class separation, where higher values indicate a more favorable feature space for the forensic task. By computing $S(l)$ for all layers $l \in \{1, \ldots, L\}$, we can quantitatively map the evolution of forensic signal strength throughout the network's depth.

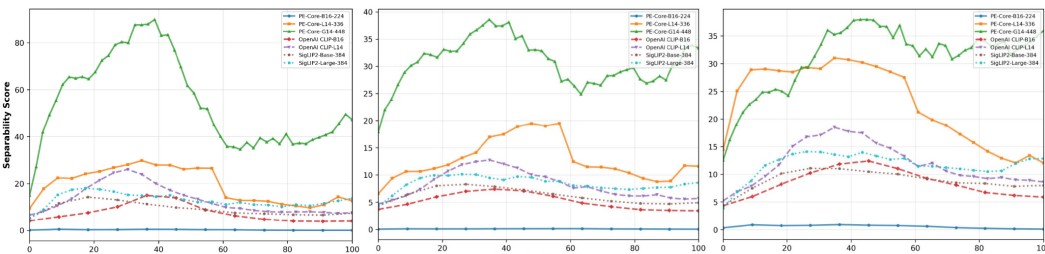

Figure 1: **Layer-wise separability analysis across 7 ViT models and 3 datasets.** The three panels display raw separability scores on different datasets: **(a)** DRCT SD1.4, **(b)** GenImage SD1.4, and **(c)** GenImage++ Flux.

## 3.2 UNVEILING THE FORENSIC SINGULARITY

**Experimental Setup for the Analysis.**    To investigate the layer-wise forensic capabilities of Vision Transformers, we conducted a comprehensive analysis across a carefully selected suite of models and datasets. Our analysis spans **seven** models from **three** major families (PE, OpenAI CLIP, and SigLIP) of varying capacities, as detailed in Appendix A.

To rigorously test the robustness and universality of our findings, we utilize three distinct real-vs-fake dataset pairs, each designed to probe for a specific type of invariance:

1. **DRCT SD1.4 Chen et al. (2024)**: Real images from MS-COCO vs. fakes from Stable Diffusion v1.4.

2. **GenImage SD1.4 Zhu et al. (2023)**: Real images from ImageNet vs. fakes from the same SDv1.4 generator.

3. **GenImage++ Flux Zhou et al. (2025)**: Real images from ImageNet vs. fakes from the more recent Flux generator.

This selection of datasets allows us to systematically evaluate the robustness of forensic features against shifts in both the **semantic domain** (DRCT vs. GenImage) and the **generative model** (Gen-Image vs. GenImage++ Flux). For our analysis, we employ the layer-wise separability score $S(l)$, as defined in Section 3, as our primary probe.

**Evidence of a Universal Representational Pattern.**    Our primary findings are presented in Figure 1, which displays the raw layer-wise separability scores for all seven models across the three aforementioned datasets. The plots reveal a strikingly consistent pattern across all high-performing models, irrespective of the dataset. We observe that forensic separability is initially low, rises steadily through the network's mid-level layers, culminates in a peak, and then declines in the deeper, more semantic layers. This characteristic inverted U-shape demonstrates remarkable stability. The pattern not only holds true when the semantic domain of real images shifts (Fig. 1a vs. Fig. 1b), but also persists when the generative model is entirely different (Fig. 1b vs. Fig. 1c). This consistent behavior strongly suggests that we are observing an intrinsic property of how ViTs represent forensic artifacts, rather than an artifact of a specific dataset.

Beyond the universal pattern, the raw scores in Fig. 1 also offer insights into the role of model architecture and training paradigms. Notably, the **PE-Core-G14 model**, with its significantly larger parameter count (2B), consistently achieves separability scores that are an order of magnitude higher than all other models, indicating that increased model capacity allows for the capture of more complex and discriminative forensic artifacts. When comparing models of a similar scale (Large-size models), **PE-Core-L14 demonstrates superior peak separability** over both OpenAI CLIP-L14 and SigLIP2-Large, suggesting its pre-training knowledge is particularly well-suited for developing strong forensic representations.

Conversely, the **PE-Core-B16 model exhibits anomalously poor performance**, with separability scores far below even other Base-sized models like OpenAI CLIP-B16. We speculate that this is a direct consequence of its training via **knowledge distillation**. The distillation process, which

typically supervises the student model based on the teacher's final-layer semantic outputs (logits), likely forces the student to develop a "semantic shortcut." This prioritizes semantic consistency at the expense of developing a rich hierarchy of low- and mid-level features. These discarded textural and structural details are, paradoxically, the very features most crucial for our forensic task.

We term the peak region of this universal pattern the **"Forensic Singularity."** It represents a critical juncture where the representation of forensic traces is maximally potent. To provide a conclusive, normalized view that distills this phenomenon, we present the aggregated cross-dataset Z-score trends in Figure 2. This plot confirms that the Forensic Singularity is consistently located between **30% and 60% of a ViT's depth**. This data-driven discovery provides a principled guideline for selecting the optimal feature source for AIGI detection and serves as the cornerstone for our proposed framework. To provide a quantitative physical grounding for this phenomenon beyond simple separability, we further probed

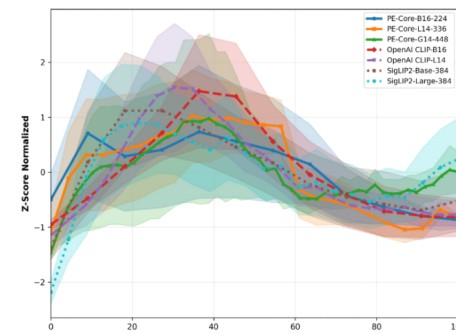

Figure 2: **Aggregated Z-score trends.** This normalized view provides conclusive evidence for the "Forensic Singularity,".

the feature space complexity via Intrinsic Dimension (ID) analysis. We constructed two probe datasets with identical semantic scope (54k images each): (1) **Pure Real**, consisting entirely of natural images from ImageNet, and (2) **Forensic Task**, a mixed manifold where 50% of images are replaced by samples from 9 diverse generators. We then estimated the ID of each layer using Maximum Likelihood Estimation (MLE).

The results, summarized in Table 1, reveal a striking dichotomy. In the Semantic Layer, the complexity remains relatively stable (only -6.3% drop), implying that generator samples provide semantic diversity comparable to real images. However, within the Singularity Layers, the introduction of generated images induces a massive **complexity drop** (~20%). This indicates that at this mid-level of abstraction, generative artifacts condense into **low-dimensional, repetitive fingerprints** that are significantly simpler than natural texture

Table 1: **Intrinsic Dimension Analysis.**

| Layer | Real | Mixed | Drop |
|---|---|---|---|
| L8 (Sing.) | 19.1 | 15.3 | **-20%** |
| L10 (Sing.) | 20.6 | 16.8 | **-18%** |
| L13 (Sing.) | 22.6 | 18.3 | **-19%** |
| L23 (Sem.) | 22.4 | 21.0 | -6% |

variations. This structural simplicity explains the "double-edged" nature of these features: while their low dimensionality yields high separability, it also makes them trivial for Euclidean classifiers to memorize (overfit), directly leading to the generalization challenge discussed next.

## 4 THE REGULARIZED HYPERBOLIC FRAMEWORK

### 4.1 THE DOUBLE-EDGED SWORD: SEPARABILITY VS. GENERALIZATION

Our Intrinsic Dimension analysis in the previous section reveals a fundamental paradox: the "Forensic Singularity" layers are characterized by **structural simplicity** (low dimensionality), not complexity. This low dimensionality is precisely what makes them a **double-edged sword**.

On one hand, the collapse of forensic artifacts into low-dimensional manifolds makes them linearly separable, explaining the high In-Domain accuracy. On the other hand, this structural simplicity makes these features trivial for a standard Euclidean classifier to **memorize**. Instead of learning a robust boundary, the model locks onto the specific low-dimensional "fingerprints" of the training generator (shortcut features), failing to generalize when the artifact distribution shifts in unseen models.

To empirically confirm this theoretical risk, we trained simple linear classifiers on individual layers of the PE-Core-L14 model using only the SD v1.4 subset. As shown in Table 2, the results validate our hypothesis. While the Singularity Layer (L13) achieves near-perfect In-Domain accuracy (99.73%), its performance collapses on out-of-domain generators (e.g., 67.63% on ADM), significantly underperforming the high-dimensional Semantic layer (74.70%).

Table 2: **Generalization performance of single-layer linear classifiers.** Models were trained exclusively on the GenImage SD v1.4 subset and tested on various unseen generators.

| PE-Core-L14 | In-Domain | Out-of-Domain Generalization | | | | | | | |
|---|---|---|---|---|---|---|---|---|---|
| Layer | SD v1.4 | BigGAN | VQDM | glide | SD v1.5 | wukong | ADM | Midjourney | OOD Avg. |
| Layer 8 | 0.9908 | 0.5083 | 0.6384 | 0.9823 | 0.9903 | 0.9065 | 0.6448 | 0.8483 | 0.7884 |
| Layer 10 | 0.9942 | 0.5983 | 0.7724 | 0.9888 | 0.9932 | 0.9513 | 0.6866 | 0.9069 | 0.8425 |
| Layer 13 | 0.9973 | 0.9576 | 0.9671 | **0.9953** | 0.9968 | 0.9879 | 0.6763 | 0.8363 | 0.9168 |
| Semantic | **0.9990** | **0.9883** | **0.9698** | 0.9755 | **0.9973** | **0.9966** | **0.7470** | **0.9069** | **0.9383** |

This creates a critical dilemma: the mid-level layers possess the strongest intrinsic signal (separability), but their low dimensionality leads to **overfitting** in Euclidean space. The final semantic layer generalizes better but lacks forensic precision. This motivates our **Regularized Hyperbolic Framework**: we need a geometry that can harness these potent, low-dimensional forensic features while preventing them from collapsing into generator-specific shortcuts.

## 4.2 GEOMETRIC INTUITION: WHY HYPERBOLIC SPACE?

The dilemma presented in Section 5.1, which is that mid-level features offer high potential but risk poor generalization, cannot be resolved by simply choosing a more powerful Euclidean classifier, as this would likely exacerbate overfitting. A more fundamental approach is needed. We begin by reframing the core nature of the task. Conventional AIGI detectors, trained as binary classifiers on both real and fake images, are often susceptible to learning spurious correlations from the real data distribution, which harms generalization Rajan & Lee (2025). This has motivated a paradigm shift towards a "real-centric" view, where the goal is to model the compact distribution of real images and detect forgeries as out-of-distribution anomalies Bi et al. (2023).

We adopt this modern perspective and formalize AI forensics as an **asymmetric anomaly detection problem**. The "real" class represents a stable "normality," while the "generated" class constitutes a vast, ever-expanding collection of anomalies. We posit that the geometry of the embedding space should explicitly reflect this asymmetry. We argue that **hyperbolic geometry**, specifically the Poincaré Ball model, provides the ideal canvas for this task.

This asymmetric problem structure demands a geometric space with corresponding properties that are remarkably well-aligned with the nature of AI forensics:

1. **Natural Center-Periphery Structure**: Unlike Euclidean space, the Poincaré Ball is inherently asymmetric, featuring a unique origin (center) and a high-capacity periphery. This provides a natural geometric parallel to the anomaly detection task. We can anchor the singular "normal" class of real images at the geometrically unique origin, while distributing the diverse "anomalous" generated classes across the vast periphery. This stands in contrast to conventional hyperbolic learning methods that map all classes to the periphery for symmetric classification.

2. **Exponential Volume Growth**: The volume of the Poincaré Ball grows exponentially with its radius. This allows the periphery to accommodate a virtually infinite diversity of present and future generative models without crowding the representation of the real class. This property is crucial for building a detector that is not only accurate today but also future-proof.

Leveraging these properties, we design a framework that operationalizes the anomaly detection philosophy within its geometric structure. Instead of learning what is "real," our model primarily learns to quantify the magnitude of generative artifacts, effectively measuring the deviation from a trusted center of normality. This provides a principled solution to the generalization challenge.

## 4.3 REGULARIZED HYPERBOLIC FRAMEWORK

Based on our geometric intuitions, we propose the **Regularized Hyperbolic Framework**, a novel framework designed to harness the power of the Forensic Singularity while ensuring generalization. The overall architecture is depicted in Figure 3. The core idea is to disentangle semantic content from forensic evidence by mapping them to the **direction** and **radius** of a point in the Poincaré Ball, respectively. The model architecture consists of four key stages.

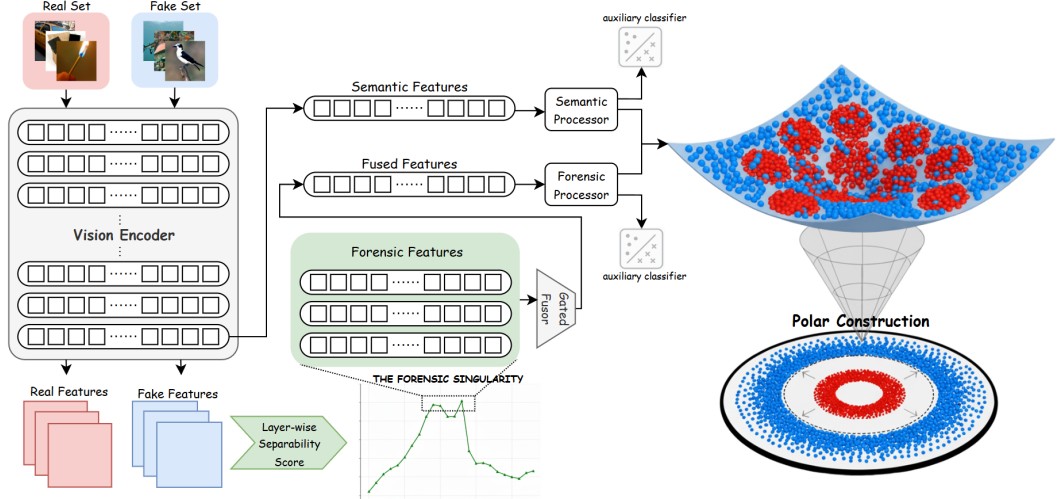

Figure 3: **Overview of Regularized Hyperbolic Framework.** We first perform a systematic layer-wise probe of a ViT backbone, which reveals a peak in forensic separability in the mid-level layers—a phenomenon we term the **"Forensic Singularity"**. Guided by this discovery, our framework extracts forensic features from the singularity region and semantic features from the final layer, processing them in parallel. These are then synergistically combined via **Polar Construction** to form a single point in the Poincaré Ball. This geometrically-informed embedding anchors the real class (red) at the origin while pushing diverse fake classes (blue) to the high-capacity periphery.

**1. Multi-Layer Forensic Feature Fusion.** To create a robust forensic representation, we first fuse features from multiple layers $\{z_{\text{for}}^{(i)}\}_{i \in \mathbb{L}_{\text{for}}}$ within the Forensic Singularity region. We employ a **Gating Feature Fusor**, $f_{\text{fusor}}$, which is an attention-based mechanism that learns to dynamically weight each layer's contribution. For each layer $i$, a gating network computes an attention weight $\gamma_i = \sigma(W_{g,i} z_{\text{for}}^{(i)} + b_{g,i})$. The final fused feature, $z_{\text{for}}^{\text{fused}}$, is then produced by a combiner network acting on the sum of the gated features: $z_{\text{for}}^{\text{fused}} = f_{\text{combiner}}(\sum_{i \in \mathbb{L}_{\text{for}}} \gamma_i \odot z_{\text{for}}^{(i)})$. This allows the model to adaptively focus on the most informative forensic traces for a given input.

**2. Parallel Feature Processing.** Next, the fused forensic feature $z_{\text{for}}^{\text{fused}}$ and the final-layer semantic feature $z_{\text{sem}}$ are processed in parallel by two dedicated MLPs: a forensic processor $f_{\text{for}}$ and a semantic processor $f_{\text{sem}}$. This step refines and projects the features into a shared $k$-dimensional Euclidean space, yielding the forensic representation $v_{\text{for}} = f_{\text{for}}(z_{\text{for}}^{\text{fused}})$ and the semantic representation $v_{\text{sem}} = f_{\text{sem}}(z_{\text{sem}})$.

**3. Polar Construction in the Poincaré Ball.** The two refined vectors are synergistically combined to form a single point $p \in \mathcal{P}^k$ in the $k$-dimensional Poincaré Ball. This is the core of our geometric regularization. The direction $u$ of the point is determined by the semantic vector, $u = v_{\text{sem}}/\|v_{\text{sem}}\|$, while its radius $r$ is determined by the magnitude of the forensic vector, $r = \tanh(\|v_{\text{for}}\|)$. The final Poincaré point is constructed as:

$$p = \tanh(\|v_{\text{for}}\|) \cdot \frac{v_{\text{sem}}}{\|v_{\text{sem}}\|} \tag{4}$$

This construction forces the model to encode forensic evidence strength as the point's distance from the origin—a powerful structural prior that aligns with our anomaly detection framing.

**4. Multi-Task Learning with Auxiliary Heads.** Finally, the classification is performed in a multi-task setup. The main branch uses a Hyperbolic Radial Classifier, $C_{\text{rad}}$, to compute logits $o_{\text{main}} = C_{\text{rad}}(p)$ based on the hyperbolic distance of $p$ to a learnable center. To guide the processors to learn discriminative, low-dimensional representations, we introduce two auxiliary linear heads, $C_{\text{aux,sem}}$ and $C_{\text{aux,for}}$, which operate directly on $v_{\text{sem}}$ and $v_{\text{for}}$ respectively. The total loss is a weighted sum of

Table 3: **Generalization performance (Accuracy %) on GenImage benchmark.** All models were trained exclusively on the GenImage SD v1.4 subset. The best and second-best results for each column are marked in **bold** and underlined, respectively.

| Model | MidJourney | SDv1.4 | SDv1.5 | ADM | GLIDE | Wukong | VQDM | BigGAN | **AVG** |
|---|---|---|---|---|---|---|---|---|---|
| Xception Chollet (2017) | 57.97 | 98.06 | 97.98 | 51.16 | 57.51 | 97.79 | 50.34 | 48.74 | 69.94 |
| CNNSpot Wang et al. (2020) | 61.25 | 98.13 | 97.54 | 51.50 | 55.13 | 93.51 | 51.83 | 51.06 | 69.99 |
| F3Net Qian et al. (2020) | 52.26 | 99.30 | 99.21 | 49.64 | 50.46 | 98.70 | 45.56 | 49.59 | 68.09 |
| GramNet Liu et al. (2020) | 63.00 | 94.19 | 94.22 | 48.69 | 46.19 | 93.79 | 49.20 | 44.71 | 66.75 |
| NPRTan et al. (2024) | 62.00 | 99.75 | 99.64 | 56.79 | 82.69 | 97.89 | 54.43 | 52.26 | 75.68 |
| SPSL Liu et al. (2021) | 56.20 | 99.50 | 99.50 | 51.00 | 67.70 | 98.40 | 49.80 | 63.70 | 73.23 |
| SRM Luo et al. (2021) | 54.10 | 99.80 | 99.80 | 49.90 | 52.80 | 99.60 | 50.00 | 51.00 | 69.63 |
| AIDE Yan et al. (2024a) | 79.38 | 99.74 | 99.76 | 78.54 | 91.82 | 98.65 | 80.26 | 66.89 | 86.88 |
| DRCT Chen et al. (2024) | **94.43** | 99.37 | 99.19 | 66.42 | 73.31 | 99.25 | 76.85 | 59.41 | 83.53 |
| OMAT Zhou et al. (2025) | 90.36 | 97.52 | 97.46 | 83.82 | 97.41 | 97.62 | 95.53 | 97.34 | 94.63 |
| **RHF/CLIP-L** | 89.59 | 98.93 | 98.64 | 73.85 | 94.27 | 98.76 | 80.50 | 68.65 | 82.94 |
| **RHF/PE-L** | 89.83 | **99.99** | **99.94** | **76.12** | **99.11** | **99.95** | **97.86** | **97.98** | **95.09** |

the binary cross-entropy (BCE) losses from each branch:

$$\mathcal{L}_{\text{total}} = \mathcal{L}_{\text{main}} + \alpha \cdot \mathcal{L}_{\text{aux}}^{\text{sem}} + \beta \cdot \mathcal{L}_{\text{aux}}^{\text{for}} \tag{5}$$

where $\mathcal{L}_{(\cdot)}$ is the BCE loss for each branch's logits, and $\alpha, \beta$ are hyperparameters. This setup acts as a regularizer, preventing overfitting to the potent but volatile features from the singularity layers.

A visualization of the learned hyperbolic embedding space, provided in Appendix C, further illustrates how our model geometrically anchors real images at the origin while pushing diverse unseen forgeries to the periphery, confirming its robust generalization mechanism.

## 5 EXPERIMENTS

We conduct a comprehensive set of experiments to validate the effectiveness of our proposed framework and the underlying principles derived from our analysis.

### 5.1 EXPERIMENTAL SETUP

**Backbones.** Our experiments are conducted on two powerful, publicly available Large-size Vision Transformer backbones: **OpenAI CLIP-L/14** Radford et al. (2021), a standard baseline in AIGI detection, and **PE-Core-L14-336** Bolya et al. (2025). As established in our analysis (Fig. 1), PE-Core-L14 exhibits a higher intrinsic forensic separability, making it a high-potential candidate. To exclusively evaluate the quality of their representations, both backbones are kept frozen during training.

**Feature Selection.** Our framework requires two types of features. The **semantic feature** is consistently extracted from the final layer of the backbone. For the **forensic feature**, guided by our discovery of the Forensic Singularity, we select a set of layers from the peak region of separability. Specifically, based on the results in Figure 2, we fuse the features from the top-3 most separable layers for each backbone. For PE-Core-L14, this corresponds to layers $\{8, 10, 13\}$, and for CLIP-L/14, this corresponds to layers $\{7, 8, 9\}$. Further details on hyperparameters and the training procedure are provided in the Appendix.

### 5.2 MAIN RESULTS AND GENERALIZATION ANALYSIS

We begin by evaluating our Regularized Hyperbolic Framework (RHF) against a comprehensive suite of prior methods on a standard set of generative models. The performance comparison is presented in Table 3.

**Performance on Standard Benchmarks.** On the in-domain test set (SD v1.4), our RHF/PE-L model achieves near-perfect accuracy (99.99%), confirming its effectiveness in learning the training distribution. More critically, in the out-of-domain generalization setting, our method demonstrates a

Table 4: **Performance (Accuracy %) on the GenImage++ benchmark.** All models were trained exclusively on the GenImage SD v1.4 subset. Column headers are abbreviated for space (e.g., "Flux Multi" for Flux Multistyle). Best and second-best results are marked in **bold** and underlined.

| Model | Flux | Flux Multi | Flux Photo | Flux Real | SD1.5 Multi | SDXL Multi | SD3 | SD3 Photo | SD3 Real | AVG |
|---|---|---|---|---|---|---|---|---|---|---|
| Xception | 36.86 | 10.48 | 4.65 | 5.45 | 97.27 | 20.63 | 38.00 | 5.83 | 15.06 | 26.03 |
| CNNSpot | 37.38 | 6.89 | 8.71 | 5.28 | 84.41 | 34.79 | 47.70 | 7.48 | 25.55 | 28.69 |
| F3Net | 25.18 | 7.79 | 2.83 | 7.90 | 94.15 | 24.01 | 46.67 | 0.84 | 30.28 | 26.63 |
| GramNet | 37.83 | 16.71 | 8.01 | 19.71 | 96.49 | 28.65 | 48.55 | 8.33 | 55.71 | 35.55 |
| NPR | 35.38 | 13.19 | 8.48 | 19.41 | 93.63 | 15.40 | 32.38 | 12.45 | 27.58 | 28.66 |
| SPSL | 67.13 | 16.55 | 43.76 | 25.73 | 71.14 | 17.74 | 44.58 | 16.22 | 29.75 | 36.96 |
| SRM | 8.46 | 2.92 | 0.37 | 1.93 | 96.62 | 6.39 | 9.97 | 0.55 | 4.43 | 14.63 |
| DRCT | 71.08 | 63.97 | 46.83 | 62.42 | 99.19 | 64.84 | 72.28 | 70.70 | 73.55 | 69.43 |
| OMAT | 96.53 | 92.55 | 97.60 | 97.67 | **100.00** | 99.17 | **98.27** | 90.38 | 98.82 | 96.78 |
| **RHF/CLIP-L** | 97.52 | 73.98 | 90.56 | 88.05 | 97.79 | 84.47 | 98.07 | 91.50 | 98.63 | 91.17 |
| **RHF/PE-L** | 95.45 | **99.44** | **99.91** | **99.85** | 97.83 | **99.68** | 97.60 | **96.63** | **99.83** | **98.47** |

Table 5: **Detailed performance (Accuracy %) on the Chameleon benchmark.** All models were trained exclusively on the GenImage SD v1.4 subset.

| | CNNSpot | GramNet | LNP | UnivFD | NPR | AIDE | OMAT | RHF/CLIP-L | RHF/PE-L |
|---|---|---|---|---|---|---|---|---|---|
| **Acc** | 60.11 | 60.95 | 55.63 | 55.62 | 58.13 | 62.60 | 66.05 | 62.20 | **94.94** |
| **F.Acc / R.Acc** | 8.86/98.63 | 17.65/93.50 | 0.57/97.01 | 74.97/41.09 | 2.43/**100.00** | 20.33/94.38 | 33.93/90.17 | 15.16/97.56 | **89.83**/98.78 |

remarkable improvement over most existing approaches. The RHF/PE-L model obtains the highest average accuracy of **95.09%** across all nine test sets, surpassing the strong OMAT baseline by a notable margin. It establishes new state-of-the-art results on five of the nine subsets, particularly on challenging generators like ADM and BigGAN, where many previous methods falter. It is noteworthy that the performance of RHF/PE-L consistently exceeds that of RHF/CLIP-L, validating our initial hypothesis from the separability analysis that PE-Core-L14 is a more potent backbone for this task. This result highlights the powerful synergy between a high-potential feature source and a geometrically-informed framework designed to harness it.

**Generalization to Unseen Advanced Generators.** To probe the limits of generalization against future generative technologies, we evaluate the models on the highly challenging GenImage++ benchmark, which features outputs from recent, advanced generators like Flux and SD3 with diverse stylizations. The results in Table 4 underscore the strength of our approach. Many prior methods suffer a catastrophic performance drop on this forward-looking benchmark. In contrast, our RHF/PE-L model demonstrates exceptional robustness, achieving the top average accuracy of **98.47%**, significantly outperforming even the powerful OMAT baseline (96.78%). It achieves the best results on five of the nine challenging subsets, showcasing its superior ability to generalize to completely unseen and stylistically diverse generative models. This result strongly suggests that our framework, by taming the potent features of the Forensic Singularity, learns more fundamental and transferable generative artifacts rather than overfitting to the patterns of a specific generator era.

**Robustness in Real-World Scenarios.** Finally, we assess the model's robustness in simulated real-world conditions using the Chameleon benchmark. This dataset is specifically designed to challenge detectors with images found "in the wild," which may include various post-processing operations and unknown generators. As shown in Table 5, when trained on SD v1.4, our RHF/PE-L model achieves a remarkable overall accuracy of **94.94%**. This result dramatically surpasses all other compared methods, including the strong OMAT baseline (66.05%), by a margin of over 28%. The detailed breakdown reveals that this is not due to a bias towards either real or fake images; our model maintains high accuracy on both (89.83% on fakes, 98.78% on reals). This outstanding performance in a challenging "in-the-wild" scenario provides strong evidence that our hyperbolic framework, by learning a geometrically principled decision boundary, is inherently more robust to the common corruptions and distribution shifts that cause conventional detectors to fail.

## 5.3 ABLATION STUDY: DISSECTING THE GEOMETRIC REGULARIZATION

To strictly validate that our performance gains stem from the proposed geometric constraints rather than simply aggregating more features, we conducted a comprehensive ablation study on the GenImage SD v1.4 subset. We investigate three critical questions: (1) Does naive fusion cause overfitting? (2) Is the hyperbolic geometry superior to Euclidean polar mappings? (3) Is the asymmetric "Real-Center" topology essential? The results are summarized in Table 6.

Table 6: **Ablation Study on Geometry and Topology.** Naive feature fusion (c) causes catastrophic overfitting. While a Euclidean Polar approach (d) recovers some performance via disentanglement, only the full Hyperbolic Framework (f) achieves state-of-the-art generalization. The Reverse-Center experiment (e) confirms that anchoring diverse Fakes to the origin destroys performance.

| Method | Feature Source | Space | Structure / Topology | OOD Avg. | $\Delta$ vs. Ours |
|---|---|---|---|---|---|
| (a) Single Layer | Semantic Only | Euclidean | Linear | 93.83% | -1.26% |
| (b) Single Layer | Singularity Only | Euclidean | Linear | 77.27% | -17.82% |
| (c) Concat-Euclidean | Sem. + Sing. | Euclidean | Linear | 77.00% | -18.09% |
| (d) Polar-Euclidean | Sem. + Sing. | Euclidean | Polar Construction | 94.08% | -1.01% |
| (e) Reverse-Center | Sem. + Sing. | Hyperbolic | Fake @ Origin | 74.49% | -20.60% |
| **(f) RHF (Ours)** | **Sem. + Sing.** | **Hyperbolic** | **Real @ Origin** | **95.09%** | - |

**The Risk of Naive Fusion (Row c).** Simply concatenating the potent Singularity features with Semantic features in Euclidean space leads to catastrophic overfitting (77.00%), performing far worse than using the Semantic layer alone (93.83%). This confirms that the low-dimensional Singularity features are prone to memorization in standard linear classifiers.

**Disentanglement vs. Geometry (Row d vs. f).** Using our Polar construction in Euclidean space (Row d) significantly improves performance to 94.08%, proving that disentangling semantic direction from forensic radius is beneficial. However, our Hyperbolic RHF (Row f) further outperforms the Euclidean Polar baseline by **+1.01%**. This indicates that the exponential expansion of the Poincaré ball provides crucial additional capacity for modeling the diverse tail of the forensic distribution, which Euclidean space lacks.

**Validation of Asymmetric Topology (Row e).** Anchoring diverse Fake images to the origin (Reverse-Center) results in the worst performance (74.49%). This empirically confirms our anomaly detection philosophy: "Real" images form a compact manifold suitable for the geometric center, while unbounded "Fake" artifacts require the extensive volume of the hyperbolic periphery.

## 6 CONCLUSION

In this work, we challenged the conventional reliance on final-layer features for AI-generated image forensics. Through a systematic, layer-wise probe of diverse Vision Transformers, we identified a universal phenomenon we term the **"Forensic Singularity"**: a consistent peak in forensic signal within the models' mid-level layers. We demonstrated that while these layers offer immense discriminative potential, they also pose a significant overfitting risk. To resolve this, we proposed a novel **Regularized Hyperbolic Framework** that tames these potent features by leveraging the unique geometry of hyperbolic space. Our framework reframes the task as an asymmetric anomaly detection problem, constructing a polar representation that disentangles semantic content from forensic evidence. This geometrically-informed design acts as a structural regularizer, promoting generalization. Extensive experiments validated our approach, showing state-of-the-art performance and, more critically, significantly superior robustness against unseen advanced generators and real-world scenarios. Ultimately, our work provides both a principled guideline for optimal feature selection in AI forensics and a powerful framework that effectively translates this insight into a state-of-the-art, generalizable detector.

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

## A  BACKBONE MODEL SPECIFICATIONS

Our systematic analysis of layer-wise separability was conducted across a diverse suite of seven pre-trained Vision Transformer (ViT) backbones, spanning three major families. This selection was made to ensure our findings on the "Forensic Singularity" are not specific to a single architecture but are broadly generalizable. All models were loaded from publicly available checkpoints, primarily from the Hugging Face Hub, and were used without any fine-tuning. Key specifications for each vision encoder are summarized in Table 7.

Table 7: **Specifications of the ViT backbones used in our layer-wise separability analysis.** All models are publicly available and were used without fine-tuning. Params indicates the parameter count of the vision tower only.

| Specification | OpenAI CLIP | SigLIP2 | Perception Encoder (PE) |
|---|---|---|---|
| **Model Size** | | | |
| *Base* | ViT-B/16 | ViT-B/16 | PE-Core-B16 |
| *Large* | ViT-L/14 | ViT-L/16 | PE-Core-L14 |
| *Giant* | – | – | PE-Core-G14 |
| **Layers** | | | |
| *Base* | 12 | 12 | 12 |
| *Large* | 24 | 24 | 24 |
| *Giant* | – | – | 50 |
| **Parameters** | | | |
| *Base* | 86M | 86M | 86M |
| *Large* | 304M | 304M | 304M |
| *Giant* | – | – | 1.8B |
| **Hidden Dim.** | | | |
| *Base* | 768 | 768 | 768 |
| *Large* | 1024 | 1024 | 1024 |
| *Giant* | – | – | 1536 |
| **Image Size** | | | |
| *Base* | 224 | 384 | 224 |
| *Large* | 224 | 384 | 336 |
| *Giant* | – | – | 448 |

The **OpenAI CLIP** models Radford et al. (2021) serve as the standard baseline in the field. The **SigLIP2** models Tschannen et al. (2025) are included as they utilize a different training objective (Sigmoid loss) compared to CLIP's contrastive loss, allowing us to test the effect of the pre-training methodology. The **Perception Encoder (PE)** models Bolya et al. (2025) are a recent family of high-performance encoders. It is worth noting that our analysis focuses exclusively on the vision tower of these models. For PE-Core-G14, this corresponds to a 50-layer, 1.8B parameter transformer. Furthermore, PE-Core-B16 is a distilled model, which provides a valuable point of comparison for the impact of training strategies on forensic feature quality.

## B  TRAINING DETAILS

All our detectors, including baselines and our proposed Regularized Hyperbolic Framework (RHF), are trained **exclusively on a single dataset**: the SD v1.4 subset from the GenImage training split Zhu et al. (2023). We do not use any data from the out-of-domain test generators during training. The training data is split into 99% for training and 1% for validation. All models are trained for a single epoch, as we observed that performance typically converges quickly on this large-scale dataset.

**RHF with PE-Core-L14 Backbone.** For our framework utilizing the PE-Core-L14 backbone, we implement a real-time feature extraction pipeline. Images are loaded and passed through a frozen PE-Core-L14 model to extract features on-the-fly. Key hyperparameters are as follows:

- **Batch Size:** 256
- **Optimizer:** AdamW with a weight decay of $1 \times 10^{-4}$.
- **Learning Rate:** A base learning rate of $5 \times 10^{-5}$ is used, with a ReduceLROnPlateau scheduler.
- **Data Augmentation:** JPEG augmentation with a probability of 0.5 is applied.

**RHF with OpenAI CLIP-L/14 Backbone.** To ensure fair and direct comparison with prior works, we also train our RHF using the widely-adopted OpenAI CLIP-L/14 backbone. The training setup is largely similar, with minor adjustments to hyperparameters for optimal performance.

- **Batch Size:** 512
- **Optimizer:** AdamW with a weight decay of $1 \times 10^{-4}$.
- **Learning Rate:** A base learning rate of $1 \times 10^{-4}$ with a Cosine Annealing scheduler.
- **Data Augmentation:** JPEG augmentation with a probability of 0.5 is applied.

## C   VISUALIZATION OF THE LEARNED HYPERBOLIC EMBEDDING

To provide further insight into the inner workings of our proposed framework, we visualize the 2D Poincaré disk representations learned by our best-performing model, RHF/PE-L. The model was trained exclusively on the GenImage SD v1.4 subset. In Figure 4, we project the feature embeddings of test images from both the in-domain generator (SDv1.4) and seven unseen, out-of-domain generators.

The visualization provides a compelling illustration of our framework's core principles and its resulting generalization capabilities:

- **Asymmetric Embedding**: The model perfectly realizes the anomaly detection philosophy by anchoring the "normal" real class (red circles) into a tight, unimodal cluster at the origin. In contrast, the diverse "anomalous" forged classes are all mapped to the high-capacity periphery.
- **Robust Generalization**: Critically, the model does not overfit to the training generator (blue triangles). Test samples from seven entirely unseen generators (colored squares) are also correctly placed outside the decision boundary. This visually confirms the quantitative results presented in our main experiments.
- **Simple and Effective Decision Boundary**: The learned decision boundary (black dashed line) is a simple circle centered at the origin. This demonstrates that our framework successfully transforms a complex, high-dimensional classification problem into a simple geometric distance check within the learned hyperbolic space.

This geometric organization is a direct result of our polar construction, where the radius is driven by the strong forensic signal from the singularity layers, effectively measuring the "degree of anomalousness" for each image.

## D   VISUALIZATION OF THE FORENSIC SINGULARITY VIA t-SNE

To intuitively validate our "Forensic Singularity" hypothesis and the findings from our Intrinsic Dimension analysis, we visualize the feature spaces of the PE-Core-L14 backbone using t-Distributed Stochastic Neighbor Embedding (t-SNE). We compare the feature distributions of the "Singularity Layers" (Layers 8, 10, and 13) against the final "Semantic Layer" (Layer 23). The visualizations are generated using a balanced subset of the GenImage dataset, comprising 54k images across 8 generative models and Real images from ImageNet.

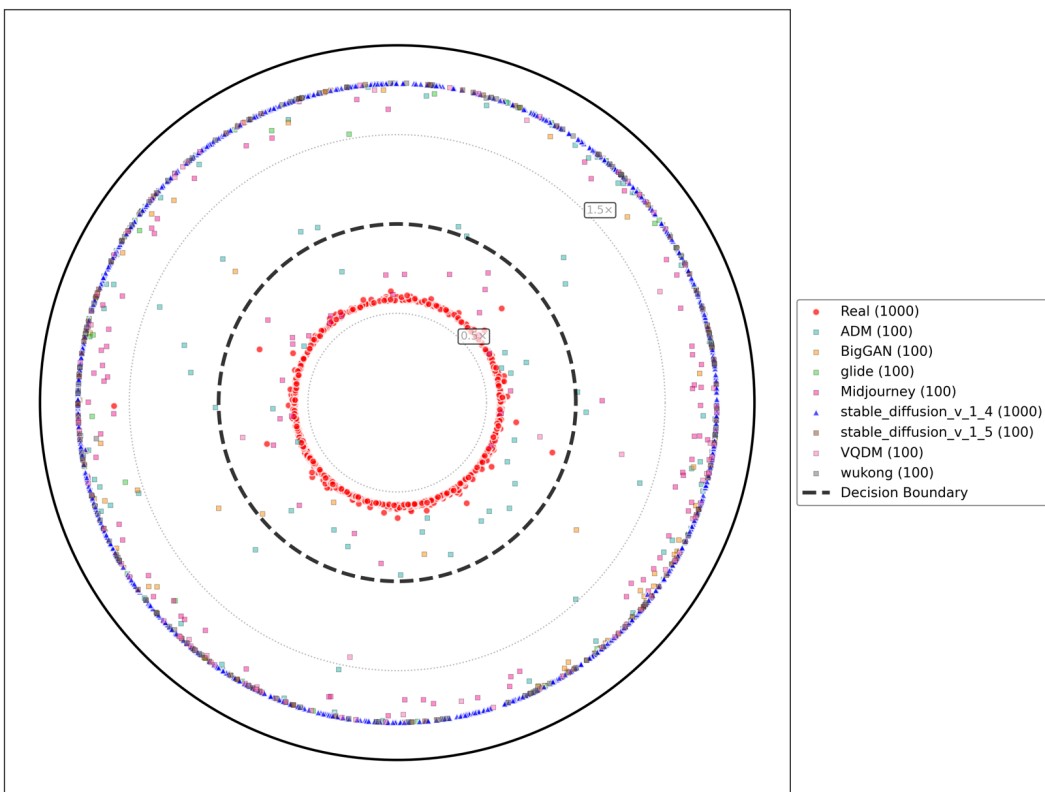

Figure 4: **Visualization of the learned hyperbolic embedding space.** We project the 2D Poincaré disk representations learned by our RHF/PE-L model, trained exclusively on SDv1.4. The model successfully learns to anchor the real images (red circles) into a compact cluster at the origin, treating them as the center of normality. All forged images, including those from the training generator (blue triangles) and **seven unseen out-of-domain generators** (colored squares), are pushed towards the high-capacity periphery. This demonstrates the core strength of our framework: it learns a simple, robust radial decision boundary that effectively generalizes to a wide variety of unseen forgery techniques by treating them as deviations from the learned real manifold.

**Analysis of Feature Distributions.** As shown in Figure 5, there is a striking structural difference between the singularity layers and the final semantic layer:

- **Singularity Layers (Layers 8, 10, 13):** In these mid-level layers (Fig. 5a-c), the features exhibit distinct, compact clustering based on the *generator source*. Real images (Red) form a cohesive cluster, clearly separated from the various fake clusters (e.g., BigGAN in Orange, ADM in Blue). This visually confirms our Intrinsic Dimension analysis: forensic artifacts manifest as low-dimensional, repetitive "fingerprints" in these layers. While this high separability provides a potent forensic signal, the distinct isolation of each generator explains the high overfitting risk observed in our Euclidean baselines—a linear classifier can easily overfit to these specific clusters, failing to generalize to unseen generators that map to different locations.
- **Semantic Layer (Layer 23):** In the final layer (Fig. 5d), the clear separation between generators vanishes. Real and Fake images are mixed together, driven by the model's objective to abstract away low-level details in favor of high-level semantic content (e.g., clustering by object class rather than image source). This confirms that relying solely on the final layer discards crucial forensic evidence.

**Justification for the Hyperbolic Framework.** These visualizations reinforce the necessity of our Regularized Hyperbolic Framework. The Singularity layers provide the necessary discrimination power (distinct clusters), but their compact, multi-manifold structure requires "taming." By anchor-

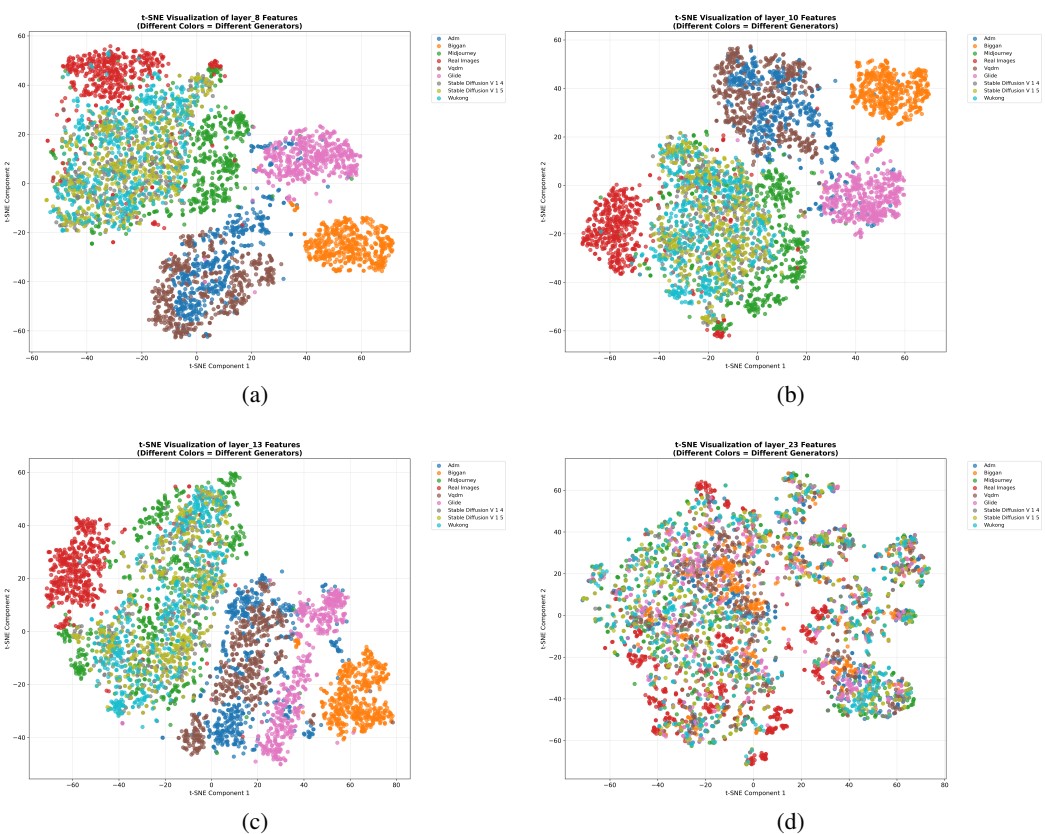

Figure 5: **t-SNE visualization of feature distributions across different layers.** Different colors represent different source generators (e.g., BigGAN, ADM, Stable Diffusion) or Real images (Red). **(a-c)** In the mid-level Singularity layers, data points cluster strongly by *generator source*, indicating rich forensic fingerprints. **(d)** In the final Semantic layer, the generator-specific clusters collapse, and Real/Fake images become mixed as the model prioritizes semantic content.

ing the Real cluster to the center of the Poincaré ball and pushing the diverse Fake clusters to the exponentially expanding periphery, our framework leverages the unique geometry of hyperbolic space to learn a generalized boundary based on "deviation from normality," rather than memorizing the specific locations of training generator clusters.

## E   ROBUSTNESS AGAINST POST-PROCESSING PERTURBATIONS

In real-world forensic scenarios, AI-generated images often undergo various post-processing operations, such as compression for social media transmission or blurring, which can erase fragile high-frequency artifacts. To evaluate the robustness of our Regularized Hyperbolic Framework (RHF), we tested our model (trained on SD v1.4 with the PE-Core-L14-336 backbone) against two common perturbations: **JPEG Compression** and **Gaussian Blur**.

We varied the intensity of these perturbations using a factor $\lambda \in \{0.95, 0.85, 0.75, 0.65\}$, where a lower $\lambda$ indicates more severe degradation (e.g., lower JPEG quality or larger blur radius).

Table 8: **Robustness Analysis under Image Degradations.** We report the accuracy (%) of the RHF model under varying intensities of JPEG compression and Gaussian blur. Even under severe degradation (Factor 0.65), the model maintains high performance ($> 83\%$), demonstrating that the learned forensic features are structurally robust.

| Perturbation Type | Intensity Factor ($\lambda$) | | | |
|---|---|---|---|---|
| | 0.95 | 0.85 | 0.75 | 0.65 |
| JPEG Compression | 88.88 | 87.29 | 89.80 | 83.56 |
| Gaussian Blur | 93.68 | 95.64 | 87.80 | 83.28 |

## F   GENERALIZATION TO A NON-CLIP BACKBONE: DINOv3

To further validate the generality and robustness of our proposed Regularized Hyperbolic Framework (RHF), we conducted an additional experiment by adapting it to a state-of-the-art, non-CLIP, self-supervised backbone: **DINOv3 (ViT-g/16)** Siméoni et al. (2025). This experiment serves to demonstrate that the core principles of our framework are not confined to vision-language models like CLIP, but are broadly applicable to other powerful Vision Transformers, even those with different training paradigms.

**Adapting RHF for DINOv3.**   DINOv3 is a versatile vision foundation model designed to produce high-quality dense features for a wide spectrum of tasks Siméoni et al. (2025). Its architecture provides a natural parallel to our feature selection strategy. The final output of its ViT encoder yields two distinct feature types that align perfectly with our framework's philosophy:

- **The [CLS] token**: This serves as a global, semantically rich representation of the entire image, which we use as the input to our *semantic processor*.
- **Patch tokens (Dense Features)**: These retain local, fine-grained textural and structural information. The exceptional quality of these dense features in DINOv3 makes them an ideal source for forensic traces, which we use as input to our *forensic processor* (after averaging).

The rest of the hyperbolic framework, including the polar construction and auxiliary heads, remains unchanged. This seamless adaptation highlights the modularity and principled design of our approach.

**Performance.**   The adapted DINOv3-based model was trained on the same GenImage SD v1.4 subset. The generalization performance on a wide range of unseen generators, including the challenging GenImage++ and Chameleon benchmarks, is presented in Table 9. The results are remarkably strong, with the model achieving near-perfect accuracy on most standard generators and maintaining high performance on newer models like Flux and SD3. Notably, it achieves an impressive

**93.66**% accuracy on the challenging Chameleon benchmark. This strong performance on a fundamentally different and more recent backbone architecture provides compelling evidence that the success of our framework stems not from a specific property of CLIP, but from its core geometric principles: the disentanglement of global (semantic) and dense (forensic) features, and the robust regularization provided by the hyperbolic space.

Table 9: **Generalization performance (Accuracy %) of our RHF adapted for the DINOv3 backbone.** The model was trained exclusively on GenImage SD v1.4. Results for GenImage++ are grouped by generator family.

| Test Generator / Benchmark | Accuracy (%) |
|---|---|
| *GenImage Subsets* | |
| BigGAN | 99.62 |
| VQDM | 99.75 |
| GLIDE | 99.18 |
| SD v1.5 | 99.71 |
| Wukong | 99.68 |
| ADM | 92.58 |
| Midjourney | 93.63 |
| **– GenImage Total Average –** | **98.00** |
| *GenImage++ Subsets* | |
| Flux | 95.28 |
| Flux photo | 99.98 |
| Flux realistic | 99.82 |
| Flux multistyle | 98.45 |
| SD3 | 98.18 |
| SD3 photo | 89.95 |
| SD3 realistic | 99.62 |
| SDXL multistyle | 98.42 |
| SDv1.5 multistyle | 96.81 |
| **– GenImage++ Total Average –** | **97.39** |
| *Real-World Scenario Benchmark* | |
| Chameleon | 93.66 |

## G REPRODUCIBILITY STATEMENT

To ensure the reproducibility of our findings, we have made significant efforts to provide comprehensive details of our methodology and experiments. All datasets used in this work are publicly available benchmarks. The specific training and testing splits, along with pre-processing steps, are detailed in Section 5.1 and Appendix B. Our proposed model, the Regularized Hyperbolic Framework (RHF), is described in detail with its geometric motivation (Section 4), architecture (Figure 3), and mathematical formulation (Section 4.3). Key hyperparameters for training are provided in Appendix B. All backbone models were loaded from the official Hugging Face Hub checkpoints, with their specifications listed in Appendix A. To facilitate direct replication of our results, we will make our complete source code, pre-trained model weights, and experiment configurations publicly available upon publication.

## H THE USE OF LARGE LANGUAGE MODELS (LLMs)

Throughout the preparation of this manuscript, Large Language Models (LLMs), specifically Gemini 2.5 Pro, were utilized as a general-purpose writing and research assistant. The primary roles of the LLM were to:

- **Refine and Polish Language:** Assisting in improving the clarity, conciseness, and grammatical correctness of the text, particularly for academic English phrasing.

- **Format LaTeX Code:** Assisting in the generation and debugging of LaTeX code for tables, figures, and mathematical equations to ensure professional and consistent formatting.

**Image Generation Model Usage (Illustration).** To create a clear conceptual visualization, we utilized an image generation model for a specific component of our main overview figure. Specifically, the visualization of the hyperbolic space and the Poincaré Disk embedding (the rightmost panel in Figure 3) was generated using **Google's Gemini 2.5 Flash (Nano Banana)**. This was done to visually represent the desired geometric distribution of real and fake data points that our framework aims to achieve.

The core research ideas, experimental design, implementation, and interpretation of results were conceived and executed by the human authors. The generative models served as tools to enhance the quality and clarity of the final manuscript's presentation, but were not contributors to the original scientific discoveries.

