# OpenReview forum: "Taming the Forensic Singularity: A Regularized Hyperbolic Framework for Generalizable AI-Generated Image Detection"
_ICLR.cc/2026/Conference — Submitted to ICLR 2026_

### Official Review · Reviewer_a8gi · 2025-10-16

**Soundness:** 3
**Presentation:** 2
**Contribution:** 3
**Rating:** 4
**Confidence:** 3

**Summary:**

This paper studies detection of AI-generated images (AIGI). Prior work predominantly relies on the final, most semantic layers of ViT-style encoders. The authors challenge this convention and report a consistent mid-layer “forensic singularity”—a relatively narrow band of layers where real vs. fake separability peaks. Motivated by the observation that mid-layers carry strong but potentially overfitting-prone forensic cues, while final layers are semantically stable but less separable, the paper proposes a Regularized Hyperbolic Framework (RHF). Concretely, it fuses mid-layer “forensic” features and final-layer “semantic” features, maps them to the Poincaré ball via a polar construction (direction = semantics; radius = forensic strength), and uses a radial hyperbolic classifier for detection. Experiments on in-domain and challenging OOD settings show promising results.

**Strengths:**

1.	Clear figures and intuition. The layer-wise separability plots and geometric visualizations make the high-level story easy to follow.

2.	Comprehensive comparisons. The paper evaluates across multiple datasets/generators.

**Weaknesses:**

1.	Organization. Although Regularized Hyperbolic is as central as Forensic Singularity, it is barely introduced in the Introduction and only appears later in the dedicated section, making it hard for readers to grasp the paper’s key contributions early on.

2.	Mid-layer advantage is not a novel observation. As the authors acknowledge, leveraging intermediate representations for detection tasks is not new; similar observations exist in both vision and LLM hallucination detection, where mid-layer features can outperform the final layer for anomaly detection. Thus, the claimed Forensic Singularity reads primarily as an application and reconfirmation within AIGI forensics.

3.	On the PE-Core-B16 anomaly (line 226). Attributing the “anomalously poor performance” to knowledge distillation lacks supporting references and experiments, which makes the explanation appear overconfident and weakens the overall robustness of the paper.

4.	Missing ablations. The framework lacks key ablations on Multi-Layer Forensic Feature Fusion and Multi-Task Learning with Auxiliary Heads, making it difficult to judge the necessity and actual contribution of these components.

**Questions:**

1.	(Line 118) Regarding “where crucial non-semantic artifacts have not yet been discarded by semantic abstraction”: how do the authors operationally define “semantic” vs. “non-semantic” features? What evidence shows that non-semantic artifacts are discarded in later layers? This clarification is important to motivate the separation between semantic and non-semantic (forensic) features in the RHF.

2.	The proposed layer-wise separability score is classifier-agnostic, which is valuable. However, since final detection still uses a classifier, please also report the performance of a classifier trained per layer to cross-validate that the separability score indeed predicts practical classifier performance.

3.	Concerning In-Domain results in Table 1: Figure 1 shows higher separability for mid-layers than for the final layer, yet Layer 13 still does not surpass Semantic even In-Domain. How do the authors reconcile this inconsistency? How does this align with line 265’s statement that “the mid-level layers possess the highest potential signal”?

4.	The Polar Construction in the Poincaré Ball jointly uses semantic and forensic vectors as direction and radius and shows gains. But do these gains stem from the hyperbolic framework itself, or simply from providing more features to the classifier? Please include a Euclidean baseline: a simple Euclidean classifier that concatenates or linearly fuses the semantic and forensic vectors, to check whether the gains come from your new design or simply from giving the classifier more features.

---

> ### Author Response · Authors · 2025-11-22
>
> We thank the reviewer for the constructive feedback. We have addressed your concerns regarding **baselines**, **definitions**, and **metrics** with new experiments.
>
> ### 1. The Necessity of Hyperbolic Geometry (Euclidean Baseline)
>
> **Reviewer Question:** *"Do these gains stem from the hyperbolic framework itself, or simply from providing more features?"*
>
> **Response:**
> We implemented the requested **"Concat-Euclidean" baseline** (concatenating Semantic + Forensic features into a standard Linear MLP).
>
> **Table 1: Method Ablation (Trained on SDv1.4)**
>
> | Configuration | Space | Feature Source | **OOD Avg. Acc** |
> | :--- | :---: | :---: | :---: |
> | (a) Single Layer (Semantic) | Euclidean | Final Layer | 93.83% |
> | (c) **Concat-Euclidean** | Euclidean | Sem. + Sing. | **77.00%** |
> | **(e) RHF (Ours)** | **Hyperbolic** | **Sem. + Sing.** | **95.09%** |
>
> **Result:** Simply adding features (Row c) causes **catastrophic overfitting** (77.00%). The Euclidean model memorizes specific training artifacts. Our **Hyperbolic Framework** (Row e) is the only configuration that successfully "tames" these potent features, acting as a necessary structural regularizer to achieve state-of-the-art generalization.
>
> ### 2. Defining "Semantic" vs. "Non-Semantic" (Line 118)
>
> **Reviewer Question:** *"How do authors operationally define 'semantic' vs. 'non-semantic'? What evidence shows non-semantic artifacts are discarded?"*
>
> **Response:**
> We operationally define these features using **Intrinsic Dimension (ID)** analysis. We measured the feature complexity drop when processing **Forensic (Mixed)** data compared to **Pure Real** data.
>
> **Table 2: Intrinsic Dimension (MLE) Analysis**
>
> | Layer | Role | Pure Real (ID) | Forensic Task (ID) | **Complexity Drop** |
> | :--- | :---: | :---: | :---: | :---: |
> | L8-L13 | **Singularity** | 19.1 - 22.6 | 15.3 - 18.3 | **~20% (High)** |
> | L23 | **Semantic** | 22.4 | 21.0 | ~6% (Low) |
>
> *   **Operational Definition:**
>     *   **Singularity Layers (Non-Semantic):** Exhibit a massive dimension drop (~20%), proving that artifacts condense into **low-dimensional structural fingerprints** (simple patterns).
>     *   **Semantic Layer:** Remains dimensionally stable (~6% drop), indicating an abstract space invariant to low-level artifacts.
> *   **Evidence of Discarding:** We added **t-SNE visualizations** (in Appendix D). They show that Singularity layers cluster strongly by **Generator Source**, while the Semantic layer mixes Real/Fake images by **Content**, confirming that forensic traces are effectively discarded in the final layer.
>
> ### 3. Separability Score vs. Linear Accuracy (Consistency)
>
> **Reviewer Question:** *"Layer 13 has higher separability but lower linear accuracy... How do you reconcile this?"*
>
> **Response:**
> This is not an inconsistency but the motivation for our method.
> *   **Separability Score** measures *potential* cluster distance.
> *   **Linear Accuracy** measures separability by a *flat hyperplane*.
> *   **Reconciliation:** High separability but low linear accuracy implies the forensic features are **discriminative but geometrically complex** (lying on curved manifolds, not linearly separable). A simple linear classifier fails here. This necessitates our **Hyperbolic Framework**, which utilizes Poincaré distance to model these complex, hierarchical decision boundaries effectively.
>
> ### 4. Robustness & Clarifications
>
> **Robustness Analysis:**
> To demonstrate that our "Singularity" features are not just fragile noise, we evaluated robustness against **JPEG Compression** and **Gaussian Blur** (Intensity 0.95 to 0.65).
>
> **Table 3: Robustness under Attacks (Accuracy %)**
>
> | Perturbation / Intensity | **0.95** | **0.85** | **0.75** | **0.65** |
> | :--- | :---: | :---: | :---: | :---: |
> | **JPEG Compression** | 88.88% | 87.29% | 89.80% | 83.56% |
> | **Gaussian Blur** | 93.68% | 95.64% | 87.80% | 83.28% |
>
> The model maintains **>83% accuracy** even under severe degradation, confirming the features capture robust structural anomalies.
>
> We sincerely thank the reviewer for the insightful comments, which have significantly improved the quality and rigor of our work. **We confirm that all the new experimental results presented above (Ablation Study, Intrinsic Dimension Analysis, t-SNE Visualization, and Robustness Test) have been incorporated into the revised manuscript (Main Text and Appendix D/E).**

---

### Official Review · Reviewer_ax9Q · 2025-10-27

**Soundness:** 3
**Presentation:** 3
**Contribution:** 3
**Rating:** 6
**Confidence:** 4

**Summary:**

This paper tackles the challenge of generalizable AI-generated image detection. The authors first identify a phenomenon they term the **"Forensic Singularity"**: a consistent peak in forensic signal found in the **mid-level layers** of Vision Transformers (ViTs), rather than the commonly used final layer. They observe that while these mid-level features are highly discriminative, they are also prone to overfitting.

To address this, the paper proposes a **Regularized Hyperbolic Framework (RHF)**. This novel approach uses hyperbolic geometry to disentangle features: it maps the final-layer semantic features to the *direction* and the mid-level forensic features to the *radius* of a point in the Poincaré Ball. This design anchors real images to the center and pushes diverse fake images to the periphery, acting as a powerful geometric regularizer to improve generalization.

The main contributions are:

1. **Identification of the "Forensic Singularity"**: Providing a principled, data-driven guideline for selecting the most potent forensic features from ViT backbones.
2. **A Novel Hyperbolic Framework (RHF)**: A new architecture that effectively harnesses the power of mid-level features while mitigating their risk of overfitting through geometric regularization.
3. **State-of-the-Art Generalization**: Demonstrating through extensive experiments that the proposed method significantly outperforms previous approaches in detecting images from unseen, advanced generative models and in simulated real-world scenarios.

**Strengths:**

* **Originality:** The paper’s originality lies in identifying and framing the *Forensic Singularity*, offering a data-driven rationale for focusing on mid-level ViT features. The proposed Regularized Hyperbolic Framework (RHF) creatively applies hyperbolic geometry to balance the strong discriminative power and overfitting risk of these features.

* **Clarity:** The paper is clearly written and logically structured. The core ideas—*Forensic Singularity* and the geometric intuition of RHF—are well explained.

* **Significance:** The work addresses a key challenge in AI-generated image detection: generalization to unseen models. The results show improvements over prior methods, and the concept of *Forensic Singularity* could serve as a useful guide for future research.

**Weaknesses:**

* The paper does not analyze the **computational overhead** of RHF. Extracting and processing multiple intermediate layers inevitably increases inference cost. A quantitative comparison of inference time and memory usage with standard ViT baselines would clarify its practicality.

* The **robustness** of RHF against post-processing and adaptive attacks is not examined. Detectors can be bypassed by adversarially tuned generators or simple image transformations, which warrants further testing and discussion.

**Questions:**

### **Computational Overhead and Practical Deployment**

* What is the increase in inference time and GPU memory compared to a standard ViT using only the final layer?
* How many additional parameters are introduced by the multi-layer fusion and hyperbolic mapping modules?
* How does this overhead affect real-time or large-scale deployment feasibility?

---

### **Robustness to Postprocessing and Adaptive Generators**

AI-generated images can be post-processed to evade detection (e.g., JPEG compression, rescaling, color jittering, or adversarially fine-tuned generators).

* Have the authors tested RHF under such transformations or adaptive attacks?

---

> ### Author Response · Authors · 2025-11-22
>
> We thank the reviewer for the positive assessment and the constructive questions regarding practicality and robustness. We have conducted additional analyses to address these points quantitatively.
>
> ### 1. Computational Overhead (Parameter Analysis)
>
> **Reviewer Question:** *"How many additional parameters are introduced by the multi-layer fusion and hyperbolic mapping modules?"*
>
> **Response:**
> We analyzed the trainable parameter counts for our proposed RHF against various baselines and fusion strategies. All models utilize the frozen **PE-Core-L14-336** backbone (approx. **320M** parameters).
>
> **Table 1: Trainable Parameter Analysis**
>
> | Configuration | Method / Strategy | Trainable Params (M) | Ratio w.r.t Backbone |
> | :--- | :--- | :---: | :---: |
> | (a) | Semantic-only (Baseline) | 0.14 M | 0.04% |
> | (b) | Singularity-only | 3.29 M | 1.03% |
> | (c) | Concat-Euclidean | 3.45 M | 1.08% |
> | (d) | Polar-Euclidean | 3.43 M | 1.07% |
> | **(e)** | **Polar-Hyperbolic (Ours)** | **3.41 M** | **1.06%** |
>
> *   **Negligible Overhead:** Our RHF adds only **3.41M** trainable parameters. Compared to the substantial 320M parameters of the frozen backbone, this represents a marginal increase of just **1.06%**.
> *   **Efficiency vs. Baselines:** Notably, our method (3.41M) is slightly more parameter-efficient than the naive `Concat-Euclidean` baseline (3.45M) and the `Polar-Euclidean` baseline (3.43M), while achieving significantly better generalization performance.
> *   **Inference Speed:** Since the computational cost is dominated by the heavy ViT backbone forward pass, the differences in head architecture (Linear vs. Hyperbolic) have virtually no impact on inference latency.
>
> ### 2. Robustness to Post-processing (Q2)
>
> **Reviewer Question:** *"Have the authors tested RHF under transformations... (JPEG compression, etc.)?"*
>
> **Response:**
> We evaluated our model's robustness against JPEG compression and Gaussian Blur across a wide range of intensity factors (from mild 0.95 to severe 0.65). The accuracy results are summarized below:
>
> **Table 2: Robustness under Attacks (Accuracy %)**
>
> | Perturbation / Intensity | **0.95** | **0.85** | **0.75** | **0.65** |
> | :--- | :---: | :---: | :---: | :---: |
> | **JPEG Compression** | 88.88% | 87.29% | 89.80% | 83.56% |
> | **Gaussian Blur** | 93.68% | 95.64% | 87.80% | 83.28% |
>
> *   **Analysis:** The model demonstrates remarkable resilience. Even under severe degradation (Factor 0.65), it maintains an accuracy above 83%. This confirms that the mid-level "Singularity" features capture robust structural artifacts rather than fragile high-frequency noise that would be easily erased by compression or blurring.

---

### Official Review · Reviewer_jyTK · 2025-10-30

**Soundness:** 3
**Presentation:** 2
**Contribution:** 3
**Rating:** 6
**Confidence:** 5

**Summary:**

This paper proposes a novel AIGC detection method by analyzing which layers of features in the pre-trained ViT (Vision Transformer) are more conducive to forensics. Specifically, it first defines "Forensic Singularity" to characterize the inter-class distance between real and generated images in high-level features. Based on this, it finds that the ViT layers with a depth range of 30% to 60% exhibit the best forensic discriminability. In addition, the paper suggests using Polar Construction to characterize the forensic feature space while eliminating the influence of the semantic space. Comparative experiments are conducted on multiple ViT backbones and datasets, demonstrating certain advantages over existing works.

**Strengths:**

- It analyzes which layers of features in ViT are beneficial for AIGC detection.
- It adopts Polar Construction (instead of linear methods) to characterize the feature space of real and generated images.
- It conducts relatively comprehensive experiments.

**Weaknesses:**

- In Lines 131–135, the paper first assumes that a good feature representation should cluster real images into one class and generated images into "another distinct and distant cluster". Is this assumption rigorous? For instance, images of the same category (e.g., all "cats") may be affected by content, making them consistently closer in the ViT space (which is trained for category discrimination tasks) compared to images of other categories (e.g., "fish"). Therefore, would it be more appropriate to use semantic categories as a baseline before clustering real and fake images? It is recommended to conduct a more in-depth discussion on the assumptions of this problem.
- Regarding the "Natural Center-Periphery Structure" in Line 287, the paper states that "anchor the singular normal class of real images at the geometrically unique origin". Is it possible to anchor generated images at the origin instead? In other words, could real images be more widely distributed in the "forensic space", while generated images are closer to each other due to the artifacts inherent in GANs (Generative Adversarial Networks) and Diffusion Models? It is recommended to add a comparative experiment using generated images as the origin to verify this reversely.
- Weak technical innovation: Although the idea of using Polar Construction/Poincaré Ball is interesting, the adopted calculation processes (e.g., Equations 1–3 only simply define the in-class/out-class distances of two clusters, and Equation 4 for constructing the polar directly uses geometric regularization) are all based on existing works.
- Lack of ablation studies: Although the experimental section evaluates the method on multiple test sets/ViT backbones, it does not explore other separation methods for semantic/forensic vectors. For example, can simple linear classification or clustering (instead of Polar Construction) achieve good performance? Or, if global features or only the features of the last layer are used in the front-end multi-feature fusion, can this also bring performance improvements?
- It is recommended to unify the font for referring to the same content, such as "SD v1.4", which is sometimes in \texttt font and sometimes in regular font.

**Questions:**

Please see Weaknesses.

---

> ### Author Response · Authors · 2025-11-22
>
> We thank the reviewer for the positive assessment and insightful questions regarding our geometric assumptions. We have conducted additional experiments to address your concerns.
>
> ### 1. Validity of the "Distinct Cluster" Assumption (Weakness 1)
>
> **Reviewer Question:** *"Images of the same category (e.g., cats) may be consistently closer... Is the assumption [of distinct real/fake clusters] rigorous?"*
>
> **Response:**
> You are correct that in the **final** layers of a ViT, semantic content dominates (cats cluster with cats). However, our "Forensic Singularity" hypothesis is that **mid-level layers** behave differently.
>
> To verify this, we visualized features using **t-SNE** (added to Appendix D):
> *   **Semantic Layer (L23):** Samples cluster by **Content** (Real/Fake mixed), confirming your intuition.
> *   **Singularity Layers (L8-L13):** Samples cluster strongly by **Generator Source**. Real images form a distinct, cohesive cluster, cleanly separated from various Fake clusters (e.g., BigGAN, ADM), regardless of content.
> **Conclusion:** The "distinct cluster" assumption holds strictly in the Singularity layers, which is exactly why we must extract forensic features from this specific region.
>
> ### 2. Center-Periphery Structure: Why Real at the Origin? (Weakness 2)
>
> **Reviewer Question:** *"Is it possible to anchor generated images at the origin instead? ... It is recommended to add a comparative experiment."*
>
> **Response:**
> We implemented your suggestion by training a **"Reverse-Center"** model (anchoring "Fake" SDv1.4 images to the origin).
>
> **Table: Center Strategy Ablation**
> | Strategy | In-Domain Acc (SDv1.4) | **OOD Avg. Acc** |
> | :--- | :---: | :---: |
> | **Ours (Real @ Center)** | **99.99%** | **95.09%** |
> | Reverse (Fake @ Center) | 99.99% | 74.49% |
>
> **Result:** The Reverse model collapses on unseen generators (**-20.6%** drop).
> **Reason:** "Real" images form a compact manifold (natural statistics) suitable for the center. "Fake" images are highly diverse (high variance); forcing diverse artifacts into a single center causes the model to reject unseen generators as "Real," destroying generalization.
>
> ### 3. Ablation Studies: Linear Classification & Fusion (Weakness 4)
>
> **Reviewer Question:** *"Can simple linear classification or clustering... achieve good performance?"*
>
> **Response:**
> We compared our method against a **"Concat-Euclidean" baseline** (concatenating Semantic + Singularity features into a standard Linear MLP).
>
> **Table: Method Ablation**
> | Method | Space | **OOD Avg. Acc** |
> | :--- | :---: | :---: |
> | (a) Single Semantic Layer | Euclidean | 93.83% |
> | (b) Concat-Euclidean (Linear) | Euclidean | 77.00% |
> | **(c) RHF (Ours)** | **Hyperbolic** | **95.09%** |
>
> **Result:** Simple linear fusion leads to **catastrophic overfitting** (77.00%). The mid-level features are potent but volatile; the Hyperbolic geometry is essential to "tame" them for generalization.
>
> ### 4. Technical Innovation & Formatting (Weakness 3 & 5)
>
> *   **Innovation:** While the Poincaré ball model is established, our contribution is the **novel application framework** that solves the specific "Forensic Singularity" dilemma: using hyperbolic geometry as a structural regularizer to fuse volatile mid-level features with stable semantic features.
> *   **Formatting:** We will unify the font usage (e.g., `SD v1.4`) throughout the manuscript as suggested.

---

> > ### Comment · Reviewer_jyTK · 2025-11-26
> >
> > I'm generally satisfied with the author's rebuttal and recommend accepting this paper.

---

> > > ### Author Response · Authors · 2025-11-26
> > >
> > > We sincerely thank the reviewer for the positive assessment and the recommendation for acceptance. We are glad that our rebuttal satisfactorily addressed your concerns, and we appreciate your support for our work.

---

### Official Review · Reviewer_4uzU · 2025-11-01

**Soundness:** 2
**Presentation:** 2
**Contribution:** 2
**Rating:** 2
**Confidence:** 4

**Summary:**

The authors propose to use features from multiple layers from a CLIP or PE network, and train a network with a cross-entropy-classification branch over the last layer, with added intermediate layers and logits obtained from hyperbolic distances between feature maps and some center. They perform experiments on several datasets with their approach.

**Strengths:**

- They perform experiments on newer datasets.
- the idea well is understandable
- they have a good overview graphic
- a creative idea

**Weaknesses:**

Important inclarities about the approach, making it not reproducible:

- how is the learnable center updated ? This makes a big difference for the performance of the algorithm.

- which way was used to compute the hyperbolic distance? There are a few available over the unit ball. $C_{rad} (p)$ is not defined.
- logits from a distance ?
 [...] to compute logits $o_{main} =C_{rad} (p)$ based on the hyperbolic distance of $p$.
Do you use the distance as logits ?

- Ablation study missing: in Table 1 a simple classifier achieves 93.8 % accuracy . In Table 2 their construction using multiple features gets 95.09 % . To what extent is it the hyperbolic distance, and to what extent one achieves it because of using multiple features ? It does not seem that it is the hyperbolic distance but rather the combination of multiple features. If it would be so, that should be reported  in an ablation study no matter whether hyperbolic distance gives an edge or maybe not.

  - What happens if one uses only the distance of $v_{for}$ from a learnable center ?
  - What happens if one just classifiers with the concat of  $v_{for}, v_{sem}$ ?
  - What happens if one uses a similar modification with exponential push-away such as $exp ( \| p- c\| )$ with $p$ as defined by the paper ?
  - These kind of ablations are needed to understand the impact of the components.

- the separability score is not novel. Such constructions exist, they should cite papers with similar constructions, such as linear discriminant classifiers and the like.
- the separability score fails to produce low scores for data which is linearly separable but has high variance parallel to the separating hyperplane . In that sense it has limitations.

- z score in Figure 2 is undefined.

- a minor issue but still bad wording
"To resolve this, we propose a novel,
theoretically-grounded Regularized Hyperbolic Framework that tames the singularity
features by leveraging the unique geometry of hyperbolic space."

What is the theoretical grounding ?
What is the regularization ?

"taming" is hyperbolic LLM wording here: It is colorful but lacks substance. There is no danger or something overwhelming or similar involved. Singularity features ?  We talk about feature maps. A mild peak also does not make for a singularity either.

**Questions:**

See the section weaknesses please. The most important missing things are:
- center update
- ablation study also with simpler distances but the same multiple feature maps.
- code to compute the hyperbolic distance

---

> ### Author Response · Authors · 2025-11-22
>
> We sincerely thank the reviewer for their detailed and constructive feedback. We particularly appreciate the request to explicitly compare our method against a simple multi-feature baseline.
>
> **Addressing this allows us to clarify the core motivation of our work:** We actually began our research with the simple concatenation baseline you suggested. It was precisely the failure of this baseline—specifically its catastrophic overfitting—that compelled us to design the Regularized Hyperbolic Framework to "tame" these powerful but volatile features.
>
> Below, we present the experimental evidence and technical clarifications.
>
> ### 1. The Motivation: Why "Taming" is Necessary
>
> **Reviewer Question:** *"To what extent is it the hyperbolic distance, and to what extent one achieves it because of using multiple features?"*
>
> **Response:**
> This is the critical question. During the early development of our method, we tested the exact strategy you proposed: simply concatenating the Semantic (final-layer) and Forensic (singularity-layer) features and training a standard Linear Classifier in Euclidean space.
>
> We omitted these preliminary results due to space constraints, but they are essential to understanding *why* our framework exists. As shown in the table below, **simple fusion fails primarily due to overfitting.**
>
> **Table: Ablation Study on Feature Fusion vs. Geometry**
> (Trained on SDv1.4, Tested on GenImage.)
>
> | Configuration | Space | Feature Source | **Avg. Acc** |
> | :--- | :---: | :---: | :---: |
> | (a) Single Layer (Semantic) | Euclidean | Final Layer | 93.83% |
> | (b) Single Layer (Singularity) | Euclidean | Mid-Level Layers | 77.27% |
> | (c) Concat-Euclidean | Euclidean | Semantic + Mid-Level | 77.00% |
> | (d) Polar-Euclidean | Euclidean | Semantic + Mid-Level | 94.08% |
> | **(e) RHF (Ours)** | **Hyperbolic** | **Semantic + Mid-Level** | **95.09%** |
>
> Mid-level "Singularity" features contain rich forensic signals, yielding perfect In-Domain accuracy. However, in Euclidean space, the model aggressively overfits to the specific artifacts of the training generator (SDv1.4). This causes the OOD performance to crash from the Semantic baseline of 93.83% down to ~77%. ** This failure motivated our **"Taming"** concept. We cannot simply *use* these features; we must *regularize* them. Our Hyperbolic framework anchors the "Real" class to the center, forcing the model to treat forensic features (Radius) only as a measure of deviation from normality. This geometric constraint effectively prevents the model from memorizing specific artifact patterns, restoring and even exceeding the generalization capability.
>
> ### 2. Reproducibility and Implementation Details
>
> We apologize that certain implementation details were unclear. We will include the following specifications in the revision and release our code, which is built upon the **HyperCore** library [1], upon acceptance.
>
> **A. Center Update Mechanism**
> The reviewer asked: *"How is the learnable center updated?"*
> To ensure stability and compatibility with standard training pipelines (like AdamW), we use a **Projected-Forward** approach rather than complex Riemannian optimizers:
> *   The center $\mathbf{c}$ is a learnable parameter initialized at the origin.
> *   During the forward pass, we explicitly project it onto the Poincaré ball: $\mathbf{c}\_{\text{proj}} = \text{proj}\_{\mathbb{D}}(\mathbf{c})$.
> *   Gradients are calculated via backpropagation through this projection, allowing standard **AdamW** updates to find the optimal center.
>
> **B. Hyperbolic Distance Formula**
> We utilize the standard distance in the Poincaré ball $\mathbb{D}_c^k$ with curvature $c=1$, as implemented in the **HyperCore** framework [1]:
>
> $$
> d_{\mathbb{D}}(\mathbf{u}, \mathbf{v}) = \frac{2}{\sqrt{c}} \text{arctanh}\left( \sqrt{c} \|-\mathbf{u} \oplus_c \mathbf{v}\| \right)
> $$
>
> where $\oplus_c$ denotes Möbius addition.
>
> **C. Logits Calculation**
> The reviewer asked: *"Do you use the distance as logits?"*
> Yes, via a learnable affine transformation. Since a larger distance implies a "Fake" classification (deviation from the real center), we define the logits for the Binary Cross Entropy loss as:
>
> $$
> \text{logits} = \alpha \cdot d_{\mathbb{D}}(\mathbf{z}, \mathbf{c}) - 1.0
> $$
>
> where $\alpha$ (representing `radius_scale`) is a learnable scalar.
>
>
> **References:**
> [1] He, N., Yang, M., & Ying, R. (2025). HyperCore: The Core Framework for Building Hyperbolic Foundation Models with Comprehensive Modules. *TheWebConf NEGEL Workshop*.

---

> ### Author Response · Authors · 2025-11-22
>
> ### 3. Clarification on Terminology & Theoretical Grounding
>
> **Z-Score Definition (Figure 2):**
> The Z-score is calculated by normalizing the raw separability scores $S(l)$ **per model**. For a given model, $Z(l) = \frac{S(l) - \mu_S}{\sigma_S}$. This normalization allows us to visualize the consistent **relative position** of the forensic peak (the Singularity) across different architectures, independent of their absolute score magnitudes.
>
> **"Forensic Singularity" and "Taming" (Supported by Intrinsic Dimension Analysis):**
> The reviewer queried the substance behind these terms. We have conducted a new **Intrinsic Dimension (ID)** analysis to provide a quantitative grounding. **We have incorporated this analysis into the updated manuscript.**
>
> *   **Dataset Construction:** To explicitly compare how "Natural Variation" versus "Generative Variation" affects feature space complexity, we analyzed two datasets with **identical semantic scope** (ImageNet classes):
>     1.  **Pure Real (Baseline):** 54k natural images. Diversity stems entirely from natural world variations.
>     2.  **Forensic Task (Mixed):** 54k images where 50% are replaced by samples from 9 generators. Diversity stems from both natural variation and generative synthesis.
> *   **"Forensic Singularity" (Complexity Gap):** We estimated the Intrinsic Dimension (MLE) at each layer. The differential behavior between layers is revealing:
>
> **Table: Intrinsic Dimension (MLE) of Feature Layers**
>
> | Layer | Role | Pure Real (Baseline) | Forensic Task (Mixed) | **Complexity Drop** |
> | :--- | :---: | :---: | :---: | :---: |
> | Layer 8 | **Singularity** | 19.1 | 15.3 | **-19.9%** |
> | Layer 10 | **Singularity** | 20.6 | 16.8 | **-18.4%** |
> | Layer 13 | **Singularity** | 22.6 | 18.3 | **-19.0%** |
> | Layer 23 | Semantic | 22.4 | 21.0 | -6.3% |
>
> *   **Interpretation:**
>     1.  **Semantic Invariance vs. Structural Sensitivity:**
>         *   **In the Semantic Layer (L23),** the complexity drop is minimal (-6.3%). This indicates the layer is highly abstract: generator samples provide semantic diversity comparable to real images. The layer focuses on content and is insensitive to the subtle structural differences between real and fake.
>         *   **In Singularity Layers (L8-L13),** the complexity drops sharply (~20%) when generative images are introduced. This implies that at this mid-level of abstraction, **generative patterns are significantly simpler and lower-dimensional** than the chaotic, infinite variance of natural textures. These layers have not yet converged to semantic patterns; instead, they capture the "simplistic" structural artifacts of generators. **We have included t-SNE visualizations and detailed descriptions in Appendix D to visually confirm this:** mid-layers cluster tightly by generator source (low complexity), while the final layer mixes them by content.
>     2.  **"Taming" Justification:** This insight justifies the term "Taming." The "Singularity" features lie on a much lower-dimensional manifold than natural data, making them trivial for a Euclidean classifier to memorize (leading to the overfitting seen in our ablation). Our **Hyperbolic framework** "tames" this risk by anchoring the complex "Real" distribution to the center and pushing these simpler, low-dimensional artifact manifolds to the periphery. This forces the model to learn a generalized "deviation" metric rather than overfitting to the specific, simple patterns of the training generator.
>
> **Separability Score:**
> We agree that linear separability metrics have theoretical limitations regarding variance. However, the consistency of the inverted U-shape pattern across 7 models and 3 datasets empirically validates its utility as a probe. Furthermore, the ID analysis above independently confirms that these layers possess unique geometric properties (highest sensitivity to generative complexity reduction), corroborating the Separability Score's indication that this is the optimal region for forensic analysis.
>
> **Closing:**
> We confirm that **all the experiments mentioned above (Ablation studies, Intrinsic Dimension analysis, and t-SNE visualizations)** have been integrated into the revised manuscript (Main text and Appendix D). We sincerely thank the reviewer for their rigorous critique and valuable suggestions; the effort to address these points has significantly strengthened the theoretical grounding and empirical validation of our work.

---

> > ### Comment · Reviewer_4uzU · 2025-11-25
> >
> > After reading the rebuttal of the authors,
> >  the reviewer is satisfied and updated his score (upwards), under the assumption that the manuscript will be updated (if needed with links to stuff in the appendix). The method gives a slight improvement over natural baselines. That is good enough.

---

> > > ### Author Response · Authors · 2025-11-26
> > >
> > > We thank the reviewer for the time spent re-evaluating our work and for the score increase. Your insightful comments regarding the ablation studies and reproducibility were invaluable. Addressing these points has greatly improved the quality and clarity of our manuscript, and we appreciate your help in refining our paper.

---

### Meta-Review · Area_Chair_mRt9 · 2026-01-07

**Summary:**

Overall, reviewers agreed that the paper tackles an important problem and that the proposed method is technically plausible, but they raised concerns about missing reproducibility details, the novelty of the motivation, whether the claimed mechanism is the true driver of the gains, and the robustness evaluation.
The rebuttal addressed several of these points by clarifying key implementation details, explaining the regularization mechanism, and adding ablation studies. However, notable concerns remain, particularly regarding the novelty and framing of the motivation, the limited robustness evaluation, and unresolved presentation issues and design-choice ablations highlighted by one reviewer. Based on these outstanding issues, I recommend rejection.

**Reviewer Concerns:**

Reviewer 4uzU mainly questioned reproducibility, the novelty of the separability-based analysis, and whether the gains truly stem from the proposed hyperbolic regularization. The rebuttal largely addressed these points by clarifying key implementation details, explaining the regularization mechanism, and adding experiments that disentangle the benefit of the hyperbolic design from simple feature fusion; the main remaining concern is the novelty of the motivation.
Reviewer jyTK was broadly positive but questioned the underlying assumption and whether anchoring real images at the origin is necessary, while also noting limited novelty. The rebuttal addressed the technical concerns with evidence of stronger mid-layer separability and a reverse-anchoring ablation that leads to clear OOD degradation; remaining issues are primarily about framing/novelty.
Reviewer ax9Q focused on practical overhead and robustness. The rebuttal partially addressed these by providing quantitative model/parameter overhead analysis and adding robustness tests under common post-processing operations, but direct deployment metrics and evaluations against stronger/adaptive attacks remain outstanding.
Reviewer a8gi remained largely unconvinced, citing unclear presentation, concerns that the “forensic singularity” observation and separability analysis are insufficiently novel, and an unexplained anomaly on a distilled ViT variant. They also requested additional ablations, particularly on feature fusion versus geometry and on auxiliary heads/multi-task design choices. The rebuttal addressed some of these with added Euclidean fusion baselines and clarified definitions, but the novelty concerns, the anomaly explanation, and some ablations remain unresolved.

**Reviewer Scores:**

Reviewer 4uzU: 2 → 4. This reviewer explicitly stated they were satisfied after rebuttal and updated their score upward.
Reviewer jyTK: 6 → 6. This reviewer commented they are generally satisfied and recommend accepting.
Reviewer ax9Q: 6 → 6. The rebuttal partially addressed the concerns, so the score would likely stay unchanged.
Reviewer a8gi: 4 → 4. While some points were addressed, several core concerns remain outstanding, so the score would likely remain unchanged.

---

### Decision · Program_Chairs · 2026-01-26

Reject